# The Unspoken Struggles from Mental Health Stigma in a Rural Community: A Qualitative Exploration of Clubhouse Members’ Lived Experiences

**DOI:** 10.3390/ijerph22111626

**Published:** 2025-10-25

**Authors:** Ruth Korankye, Gloria Oladeji, Lauren Gilbert

**Affiliations:** 1Division of Kinesiology and Health Promotion, University of Wyoming, Laramie, WY 82071, USA; goladeji@uwyo.edu; 2Department of Human Development and Family Sciences, University of Connecticut, Storrs, CT 06269, USA; 3Rural Health Institute, University of Wyoming, Laramie, WY 87071, USA; lgilbe10@uwyo.edu

**Keywords:** clubhouse model, mental illness, mental health stigma, rural mental health

## Abstract

Rural communities have a close-knit social structure, hindering the disclosure of mental disorders due to fear of negative societal perception. The study aimed to explore the experiences of both clubhouse members and staff with stigma and to examine how the Clubhouse addresses stigma in rural Wyoming through semi-structured interviews. Semi-structured interviews were conducted with 16 participants (12 clubhouse members and 4 staff members). The data analysis was performed in NVivo using reflexive thematic analysis. Three main themes were generated: “mental health prejudices,” where participants reported being perceived as dangerous, unintelligent, incompetent, and attention seekers. The second main theme, “the root causes of mental health stigma,” has three subthemes: “mental health illiteracy”, “the media,” and “personal struggles and background.” The third main theme, “clubhouse effort to address stigma,” encompasses two subthemes: “the unique clubhouse environment for self-stigma recovery” and “advocacy and community outreach.” The study findings highlight the emotional challenges individuals with mental illness face due to stigma from the media and the public. However, the clubhouse provides a non-judgmental environment that addresses both self- and public stigma. The findings also support expanding clubhouses, especially in rural areas, to improve mental health outcomes.

## 1. Introduction

Globally, mental health disorders are the leading cause of years lived with disability (YLDs), contributing to the public health burden [1]. In the United States, approximately 23.1% of adults live with a mental disorder [2]. Geographically, the prevalence of mental disorders in urban areas is consistent with that of rural areas [3,4]. However, rural residents experience a higher risk of suicide compared to those in urban areas, often due to limited mental health resources, including healthcare providers and mental health facilities, compared to metropolitan areas [4,5,6]. Beyond the limited mental health resources, individuals with mental disorders often delay seeking medical care because of limited mental health knowledge, stigma, and discrimination [3,7].

Mental health stigma is defined as negative beliefs, attitudes, and stereotypes targeted at individuals with mental disorders [8]. The stigma can be either self-stigma, which involves internalized negative beliefs and blaming oneself for one’s condition, or public stigma, which is a negative attitude and beliefs from individuals towards those who live with mental illness [8]. Most rural communities have a close-knit social structure, hindering the disclosure of mental disorders or seeking help due to fear of negative perception from their family and society, as mental disorders are perceived as a sign of inefficiency [6]. This fear delays seeking early healthcare, resulting in worsening of mental condition, emergence of other medical conditions (heart disease, diabetes, and obesity), decreased quality of life, and eventually increased mortality [9]. The effect of mental health stigma extends beyond affected individuals to their families and society. Families often experience guilt and embarrassment due to the stigma associated with mental health, which can lead to isolation, depriving these individuals of the emotional support necessary for recovery [9,10]. Stigma also places society at a disadvantage in the allocation of mental health resources, as mental health services are often under-resourced [7,9].

Mental disorders incidence has been predicted to decline over the next 20 years, yet the prevalence of the condition is estimated to increase due to increased population and societal pressures [1]. This projection calls for effective mental health management strategies, particularly addressing stigma, a primary contributor to delayed health-seeking behavior [1]. Addressing stigma can improve the mental health of affected individuals, motivating their families and society to provide the emotional and social support needed for recovery [9]. Research indicates that as self-stigma increases, recovery is less likely to occur [11]. Most policymakers and mental health institutions propose anti-stigma interventions, including peaceful protest, educational campaigns, social contact, legislative reform, advocacy, and stigma self-management, to protect affected individuals with mental disorders [8,12]. Despite these strategies being in place, stigma is still on the rise in rural communities due to the close-knit structure of these areas.

Community-based services such as the Clubhouse model help implement these anti-stigma interventions effectively. The Clubhouse model is a community-based service established to support and empower individuals with mental disorders. The organization’s objective is to promote self-esteem among its members and offer opportunities, including employment, socialization, education, skill development, housing, and improved wellness [13]. Research shows that the clubhouse plays a significant role in addressing mental health stigma. For instance, a previous study found that clubhouse members perceive their clubhouse as a “stigma-free zone”, which motivates them to develop self-love and a positive mindset [14]. Further research suggests that stigma is perceived less in clubhouses compared to rehabilitation training centers [15]. Despite the effectiveness of the clubhouse model in reducing mental health stigma, they are limited to rural areas. Wyoming, one of the most rural states in the United States, has a population density of approximately six people per square mile [16]. Residents are widely dispersed across the state, with a small population in each town, making it difficult for one to disclose or seek early mental healthcare [6,16]. Despite a population of 587,618 residents in the state, there is only one clubhouse that serves 23 counties [17]. The state also experiences a limited number of mental health professionals, despite being burdened with a high prevalence of mental disorders and an elevated suicide rate [18]. In 2022, 155 deaths were attributed to suicide and increased slightly to 157 in 2023, corresponding to an age-adjusted rate of 26.3 per 100,000 people, ranking Wyoming as the state with the third-highest suicide rate [19,20]. These findings highlight the need to establish a more psychosocial environment, such as clubhouses, to address this public health burden in the state.

Several frameworks situate the research questions. First, a recovery-oriented framework guided the initial research, recognizing that recovery is a unique journey for each individual and is not focused solely on the eradication of symptoms, deficits, or illness [21,22]. This framework focuses on hope, personal agency, the social environment, and social inclusion as major components of the recovery journey [22]. This framework aligns with the Clubhouse model, which also focuses on member choice, collaboration, and purposeful activity [13]. This framework shaped the project’s broader scope to better understand recovery experiences, with individuals living empowered, hopeful, and satisfying lives even amid ongoing mental challenges.

A stigma framework was also utilized to contextualize recovery and the multiple levels of stigma within the Clubhouse Model. The individual (self-stigma), interpersonal (public stigma), and structural (policies and practices) levels of stigma can limit opportunities for employment, education, and social participation [23,24]. This framework provides a lens for examining how Clubhouse membership can help alleviate stigma and buffer against it while promoting inclusion and equity.

This project also used a community-engaged research framework to guide its work and address the community’s needs. Our community partner wanted the voices of their clubhouse members to be heard in the larger discussions about mental health in their community. We worked with the community partner to develop the project, including the interview question guides, the incentive payment amounts, and the recruitment process. We continue to engage them in the data analysis and interpretation phase, as well as in disseminating the findings and outlining the organization’s next steps.

The purpose of this study was to explore the experiences of both clubhouse members and staff with stigma and to examine how the Clubhouse addresses stigma in rural Wyoming, using semi-structured interviews. This study was designed to answer the following research questions: (1) How do clubhouse members perceive the origins of mental health stigma? (2) How do clubhouse members and staff describe their personal and professional experiences with mental health stigma? (3) What are the strategies the clubhouse model employs to address mental health stigma in rural Wyoming?

The semi-structured interviews were tailored to each group: members’ questions focused on their personal experiences with stigma, while staff focused on the Clubhouse’s mission and member support.

## 2. Materials and Methods

### 2.1. Study Design

A phenomenological qualitative study design was employed to explore the experiences of both the clubhouse staff and members regarding stigma and the clubhouse’s role in addressing it. This design was chosen because it focuses on capturing the essence of participants’ lived experiences, allowing for an in-depth understanding of how individuals experience mental health stigma within rural communities [25]. By using phenomenology, the study aimed to develop a detailed description of staff and members’ perspectives, highlighting the clubhouse’s role in supporting individuals with mental health conditions in rural areas.

### 2.2. Study’s Setting and Participants

The study was conducted in collaboration with the Clubhouse in Wyoming, the only clubhouse serving individuals with mental illness across the entire state. Clubhouse members are typically referred to by a healthcare professional or introduced by a friend or relative through word of mouth.

Participants were recruited using purposive sampling between August and September 2024. A total of 16 participants were included in the study, comprising 12 clubhouse members and 4 staff members. Recruitment concluded after 16 participants, as no additional volunteers were found. Additionally, subsequent interviews after our 10th interview revealed similar findings, indicating that data saturation had been reached. Based on this, the research team determined that sufficient data had been collected to support the exploration of the study objectives. Recruitment was conducted through flyers and verbal invitations by the researchers and Clubhouse staff, including the director. Eligibility criteria included being a current member or staff member and being at least 18 years old. The participants were predominantly female (*n* = 12), with four males. Most clubhouse members had an intermediate membership, defined as involvement with the clubhouse for 6 months to 5 years.

### 2.3. Ethical Considerations

The study received approval from the University of Wyoming Institutional Review Board (IRB Number: IRB-2024-228). All participants provided written informed consent before participating in the study, and pseudonyms were assigned to ensure confidentiality. As part of informed consent, participants were informed that they could withdraw from the study at any time without consequences.

### 2.4. Data Collection

A recruitment flyer was distributed throughout the clubhouse with information about the study and instructions on how to participate. The flyer included a Qualtrics link to collect participants’ demographics and available interview times. Interested participants were contacted to schedule an interview either in person or online using a professional university Zoom account.

A semi-structured interview guide was developed based on the literature review and in collaboration with the Clubhouse’s director. Members were asked about their experiences, perceptions, and behaviors within the context of their Clubhouse membership. Staff interviews followed a similar structure but excluded questions about personal experiences with mental health and discrimination, focusing instead on the Clubhouse’s mission and the support it provides, and on how those goals are achieved. The interviews lasted between 15 and 75 min, with variability largely reflecting participants’ availability and the length oftheir experience with the Clubhouse environment. Newer members had limited experience and could not speak to the Clubhouse’s services. Long-term members spoke at length about the history of the clubhouse, the evolution of services, and changing community perspectives. Participants received $100 gift cards for their participation in the study. This amount was agreed upon with the community partner organization as an acceptable incentive for their participation in the community-engaged research process. All interviews were recorded and later transcribed either by a professional third-party company for in-person interviews or via Zoom’s auto-transcription features. Transcripts were subsequently double-checked and formatted by trained researcher assistants before analysis.

### 2.5. Data Analysis

Reflexive thematic analysis was employed to analyze transcripts of both the clubhouse members and staff. The data were exported to NVivo (Lumivero, Denver, CO, USA), a qualitative data analysis software [26]. The analysis followed a six-phase approach for reflexive thematic analysis [27]. In the first phase, the two researchers familiarized themselves with the data during the transcription and cleaning process, followed by repeated readings prior to analysis. In the second phase, both researchers independently generated initial codes from the transcripts to capture meaningful patterns. Discrepancies in initial codes were resolved through team discussions led by the study supervisor, resulting in consensus and consistent interpretation among researchers. During the third and fourth phases, the researchers collaboratively reviewed and refined the codes to develop preliminary themes, ensuring that each theme accurately represented the data and aligned with the study’s objectives. In the fifth phase, the themes and subthemes were clearly defined and named to reflect their essence. Finally, in the sixth phase, the researchers synthesized all the themes to gain a clear understanding of the data and present the findings. An initial review of the data revealed that all perspectives were more effectively analyzed when considered inclusively, rather than examined separately from the staff.

### 2.6. Researcher Team Positionality and Reflexivity

Our study team members’ intersecting identities and shared commitment to reducing health inequities for rural populations affected study design, interactions with study participants, and data analysis. Our perspectives include intersecting racial and ethnic identities (African American, Ghanaian, and Nigerian), research experiences (career researcher and graduate students), and disciplines (sociology, public health, clinical medicine, food system and nutrition, community health, and mental and behavioral health). Both RK and GO are new to mental health research, which helped minimize potential bias during data interpretation. Throughout the analytical process, the researchers held regular meetings and engaged in reflective discussions to examine how individual research backgrounds might influence data interpretation. Additionally, we consulted with our community partner on our data analysis and interpretation.

### 2.7. Trustworthiness

Several strategies were employed to ensure the trustworthiness of the study findings. Three forms of trustworthiness were employed: researcher triangulation [28], persistent observation, and reflexivity [29,30]. For researcher triangulation, multiple researchers with diverse areas of expertise were involved in the conceptualization, analysis, and completion of the study. Persistent observation was maintained throughout data collection and analysis, and frequent meetings were held during the data analysis phase. To ensure reflexivity, each researcher actively reflected on their experiences and preconceptions, both individually and at group discussions, to acknowledge any researcher bias and minimize its influence on data interpretation.

## 3. Results

The interview analysis generated three main themes: mental health prejudices, the root causes of stigma, and the clubhouse’s efforts to address stigma, with some themes having two or more subthemes that further illustrate and support the overarching themes.

### 3.1. Mental Health Prejudices

Participants described a range of negative beliefs held by the public about mental health. The most frequently mentioned were that individuals with mental illness are dangerous, incompetent, unintelligent, and attention seekers. These assumptions often result in fear, judgment, and avoidance towards those with mental disorders. Stevie, a clubhouse member, highlights: “That the person with the mental illness is high risk or flight risk for fighting with people physically, or just maybe feeling uncomfortable with being with that person in the same room or place.” Abby, a clubhouse staff member, also shared her perspectives on the public’s misconceptions of people with mental illness: “[people with mental illness are] weak, they won’t show up for work, you know. They can’t even make it through a whole day.”

Participants also shared their personal experiences. Libby, a clubhouse member, recounted losing loved ones after disclosing their condition: “I’ve lost a lot of relationships over my mental illness, like people who just learned about my mental illness, and they just didn’t want to deal with it.” Eve, a clubhouse member, shared how she almost lost her life when she was traveling because of this discrimination:


*I was feeling really weird and stuff, and I couldn’t understand why. So, they did—Is there a doctor on the plane?. Anyway, the guy came up and he said he was in internal medicine. He looked at my medication and said, ‘She’s just faking it for attention. She’s on psych meds.’ Found out, you know what it was? My oxygen disconnected from the machine.*


### 3.2. The Root Causes of Stigma

#### 3.2.1. Mental Health Illiteracy

Most of the members identified a lack of understanding and education about mental health as the primary cause of stigma. They emphasize that the public often lacks enough information about mental disorders, which contributes to negative perceptions of them. Stevie, a clubhouse member, shared: “I don’t think they do it to inflict harm upon the person that has a mental illness, but I do feel as though maybe they are uneducated about it.”

#### 3.2.2. The Media

Another source of stigma that was reiterated was the media. Most of the participants complained of misrepresentation and negative portrayals of mental disorders in our media space. Media representation often depicts people with mental disorders as dangerous, fostering fear in their viewers, resulting in the avoidance of individuals with mental disorders. Furthermore, the media exaggerates the traits of characters with mental health conditions, portraying them as overly intelligent, which misrepresents the reality of living with such conditions. Sandra, a staff member, described a scenario as:


*I think maybe part of it might be like the TV has made it where maybe they’re afraid that they’re just going to go crazy and kill everybody, or hit them, or start yelling at them, or even maybe a fear of not knowing how to talk to them.*


#### 3.2.3. Personal Struggles and Background

Stigma was also attributed to the public’s internalized struggles, where their frustrations are often projected onto those with mental disorders. Several participants shared their opinions that emotional suppression over time can lead to exhibiting negative attitudes towards those with mental disorders. Amber, a clubhouse member, said: “Well, honestly, in my opinion, there’s something going on in their life that they’re struggling with, and they have to find a way to make someone else hurt because they hurt.”

Additionally, other participants ascribed that one’s upbringing and sociocultural background can influence a stigmatizing attitude. Cultural norms and family beliefs were seen as contributing factors. Stacey, a staff member, described:


*I think that stigmas probably go back to the beginning of time, but we can just start with going back one generation. Let’s go back two generations, the Boomers. They were raised by people who survived the Great Depression, suffered a lot of poverty, and I think all the phrases about ‘pulling yourself up by your bootstraps,’ Boomers doing that. When they had their kids—that would be Gen X—they were still pulling themselves up by their bootstraps, working all the time.*


### 3.3. Clubhouse Effort to Address Stigma

Mental health stigma can manifest as self-stigma or public stigma. The clubhouse model addresses both forms of stigma through two subthemes: the unique clubhouse environment for self-stigma recovery and advocacy, and community outreach.

#### 3.3.1. The Unique Clubhouse Environment for Self-Stigma Recovery

The clubhouse was described as a unique and welcoming atmosphere for individuals with mental disorders, helping to alleviate self-stigma. Several participants noted that they were treated with respect and dignity, unlike in other environments. Libby, a clubhouse member, shared: “I think that my favorite part of being part of the clubhouse is that everybody treats you with respect, and you get treated like a human being instead of somebody who has a mental illness.” The clubhouse members emphasized that they were treated equally, irrespective of their diagnosis. Together, these experiences help restore their self-worth. Other participants highlighted that the clubhouse’s non-judgmental structure creates a safe place to visit regularly. Emily, a clubhouse member, expressed this sentiment:


*I found it’s no judgment, really it’s just a judgment-free zone where life can be really judgy that’s my main experience. They don’t judge. They look at you, and you feel like, I don’t know that I can do this and they are like, yeah, you can, you got it. We’ll help you. If you can’t, we’ll teach you. It’s okay not to understand everything.*


Participants also reported that emotional support in the clubhouse helps alleviate self-stigma. Since stigma often hinders individuals from seeking or receiving emotional support, the clubhouse was seen as a refuge during emotional distress. Amber, a clubhouse member, stated:


*They’ve been a really good support, really good backbone for me. Honestly, like anytime I have moments, and I need them, they’re there. So instead of having to call 988 or even the ambulance to come pick me up to take me to the clinic, I can call them and they’ll guide me through a moment, and that, they’re really good support in my opinion.*


#### 3.3.2. Advocacy and Community Outreach

The clubhouse plays a vital role in addressing public stigma at the community level through advocacy and community engagement. They organize community fundraisers to support the clubhouse. This also serves as a platform to educate the public about mental health, highlight the clubhouse’s impact, and offer opportunities for members and staff to share their experiences. This act instills a positive mindset in the people and encourages those with mental health diagnoses to seek psychosocial support at the clubhouse. Angie, a clubhouse member, shared: “they advocate for the people that go there for services, and I see that. I want to help too, and I want to be a help and not a hindrance. “

In addition to public events, the Clubhouse collaborates with community stakeholders, including law enforcement agencies. They offer guidance on how to manage situations involving individuals with mental disorders. Furthermore, they partner with organizations to ensure access to essential resources such as employment and housing, which can be particularly difficult for those with mental disorders to obtain. As Sandra, a staff member, explained:


*We’re trying to work so closely within the law enforcement site we have when they do their CIT [Crisis Intervention Team] training. We’re trying to have them into the clubhouse where members are speaking to them and giving them their experiences better, good, you know, with the cops, I think that’s a huge place that needs help, too, is being able to for the police officers to know how to handle a situation with somebody that’s mentally ill.*


## 4. Discussion

This study explored the clubhouse members’ and staff’s experiences regarding mental health stigma, as well as strategies used by the clubhouse organization to address these stigmas. The results revealed three primary themes: (1) mental health prejudices, (2) the root causes of stigma, and (3) the clubhouse’s effort to address stigma. These findings highlight the significant impact of stigma on individuals with mental disorders and emphasize the essential role of the clubhouse in reducing stigma and supporting their mental well-being.

### 4.1. Mental Health Prejudice

Many individuals living with mental disorders experience preconceived notions and negative stereotypes about their condition, and the study participants were no exception. The most frequent misconception about mental disorders is that affected individuals are dangerous, and people mostly avoid them out of fear of being harmed. Our study revealed similar assumptions that those with mental illness are seen as threatening and incompetent. These findings align with existing literature [31,32,33]. Surprisingly, other misconceptions were that individuals with mental disorders are unintelligent and attention seekers; for instance, a participant almost lost her life due to the perception of being an attention seeker. Individuals with mental disorders often lose emotional support from their families due to the stigma associated with being linked to those affected [10]. Furthermore, those with mental disorders are often seen as incapable, unreliable, and unproductive, which results in difficulty finding employment and economic hardship [34]. While exploring mental illness in the labor market, one study revealed that individuals with mental disorders received fewer callbacks from the hiring agency than those with only physical illness, reflecting employers’ discrimination against those with mental disorders [35]. These mental health prejudices exacerbate the struggles these individuals face, making life unbearable.

### 4.2. The Root Causes of Stigma

The study identified mental health illiteracy, the media, personal struggles, and background as the primary contributors to stigma. Among these, nearly all participants emphasized mental health illiteracy as the major cause. This finding aligns with existing literature, which suggests that a lack of understanding and awareness fosters negative perceptions of mental illness [36,37]. Despite the rise in anti-stigma initiatives that focus on public advocacy and education, participants believe that the public lacks adequate knowledge of mental health conditions. This perception is confirmed by a systematic review and meta-analysis, which found that although mental health literacy has improved over time, stigma persists [33]. In rural communities, limited access to mental health resources and the socially close-knit nature of these areas contribute to lower levels of awareness compared to urban settings. This often discourages individuals from disclosing their conditions, making it harder for the community to recognize the prevalence and seriousness of mental illness [9,38]. Several studies recommend engaging community leaders and addressing cultural norms in mental health awareness campaigns to influence perceptions and reduce stigma [38,39,40].

The media was also identified as a primary contributor to mental health stigma. Participants expressed concern about the portrayal of individuals with mental disorders as dangerous, incompetent, or unintelligent in various media outlets. These portrayals are consistent with previous studies, which suggest that the media often misrepresents mental disorders by associating them with violence, terrorism, and crime. This misrepresentation contributes to the prevalent stereotype that individuals with mental disorders are dangerous [41,42,43]. In reality, a few instances of violence are associated with mental disorders. However, due to extensive misinformation, many violent incidents are wrongly attributed to mental disorders without concrete evidence, fostering public fear and the social isolation of affected individuals [41]. Given the media’s global reach and influence, it can serve as an ideal platform for mental health advocacy. Accurate reporting, inclusion of narratives from individuals with lived experiences, and liable coverage, particularly in refraining from attributing violent acts to mental illness without confirmed diagnoses, could contribute meaningfully to the reduction in stigma [41].

Personal struggles and background were identified as contributing factors to mental health stigma. Both staff and members noted that stigma often stems from individuals’ internalized struggles and long-standing suppressed emotions, which can lead them to project their frustrations onto those with mental health conditions. Cultural beliefs and upbringing also play a significant role in shaping attitudes toward mental illness. In some cultures, mental illness is viewed as a punishment for wrongdoing, while others perceive it as a sign of personal weakness, where individuals are expected to exercise self-restraint [6,44]. Growing up in such environments can cause individuals to internalize these stigmatizing beliefs, making them less likely to understand or empathize with those experiencing mental health challenges.

### 4.3. Clubhouse’s Effort to Address Stigma

The participants emphasized the importance of the clubhouse in addressing both self-stigma and public stigma at the community level. In addressing self-stigma, the clubhouse was described as a unique environment that fosters recovery from self-stigma. Several participants shared that they were treated with respect and dignity, within a no-judgment environment, which made them feel like valued individuals rather than being labelled mentally challenged. This finding is consistent with previous clubhouse studies, which reveal how members regain self-esteem through participation [14,16].

Due to persistent societal perceptions that portray individuals with mental illness as incapable, weak, or unintelligent, or attribute their conditions to spiritual causes, many internalize these beliefs and gradually lose their confidence and self-worth [45]. However, the clubhouse model promotes a culture of equality, dignity, and mutual respect, regardless of diagnosis. Beyond promoting self-worth, a previous study found that the clubhouse provides competence, relatedness, and autonomy—the three basic psychological needs outlined in Self-Determination Theory (SDT)—which facilitates overcoming self-stigma and promoting psychological well-being [46].

In addressing public stigma, the clubhouse plays an advocacy and community outreach role, focusing on mitigating stigma at the community level, where stigma often originates. Participants expressed that the clubhouse is recognized for advocating for mental health, particularly through fundraising, an ideal platform for fostering connections between individuals with and without mental disorders. The clubhouse has implemented various anti-stigma strategies, including educational campaigns, social contact, advocacy, and self-management of stigma, all of which are facilitated through this platform. The community members encounter individuals with mental health conditions, which not only introduces them to the clubhouse but also connects them with others facing similar diagnoses.

Though educational campaigns are important, many studies recommend integrating education with direct social contact, as negative public attitudes toward mental illness often persist despite constant education [33,47]. A key limitation of many educational campaigns is their focus on the biogenetic and neurological explanations of mental illness, rather than emphasizing the importance of treating individuals equally [48]. Social contact, on the other hand, can be an effective tool for changing perceptions, helping mitigate stigma by positively influencing the public perception of harmful stereotypes that individuals with mental disorders are dangerous, incompetent, or unintelligent [47,49].

These results also reinforce the Clubhouse model’s ability to directly address stigma at different levels by creating spaces where members are valued for their contributions and capabilities rather than their diagnoses. Self-stigma is challenged through members’ participation in meaningful work and the relationships they build within the Clubhouse, thereby cultivating a sense of agency and belonging. Stigma at a public level, such as narratives of incapacity and dependence, is also combatted as members are actively engaged in their community and in different work and school settings. Furthermore, the Clubhouse offers pathways to mitigate structural stigma through opportunities for employment, education, and community engagement. The Clubhouse model offers a recovery-oriented space for individuals in rural and underserved areas to thrive through empowerment and inclusion in their personal journeys of recovery while actively disrupting stigma in their communities.

### 4.4. Limitations

The small sample size of the current study limits the representativeness of both the clubhouse members’ and staff’s perspectives. However, these findings provide a novel perspective on Clubhouse membership in a rural community.

Although some interviews were shorter than anticipated, they provided focused and relevant insights. Because this was a new research initiative, some participants may have also felt hesitant to share more in-depth experiences until greater trust was established with the research team. However, the data were sufficient to achieve thematic saturation across participants, and triangulation with field notes enhanced the credibility and richness of the findings.

### 4.5. Study Implications

This study provides valuable insights into the role of community-based models, such as the clubhouse, in addressing mental health stigma, particularly at both the individual and community levels. The findings underscore the importance of psychosocial environments that enhance self-esteem and emotional well-being for individuals living with mental health conditions, particularly in rural communities where stigma is heightened.

Mental health professionals and community organizations, particularly those in rural areas, can use these findings by incorporating social contact and engaging community leaders, such as pastors, teachers, and elders, in educational campaigns and outreach efforts. Such strategies may enhance the effectiveness of anti-stigma initiatives, as residents in rural communities often hold their leaders in high esteem and place trust in their guidance.

Additionally, this study offers important implications for policymakers. It highlights the need for policymakers to implement and expand community-based services, like the clubhouse model, to address the limited availability of mental health resources in rural settings. Integrating the clubhouse model within rural health systems could help bridge service gaps, enhance care coordination, and improve accessibility for residents who are geographically dispersed.

Future research should explore the impact of clubhouse programs in other rural regions across the United States. Given the limited literature and smaller sample size, further investigation is needed to understand how this model reduces stigma, strengthens social networks, and promotes recovery in resource-limited rural communities.

## 5. Conclusions

Mental health stigma significantly impacts the lives of individuals with serious mental illness and disorders, affecting their emotional support systems, economic stability, and health-seeking behaviors. These stigmas are largely driven by mental health illiteracy, media portrayals, and the public’s internalized struggles and cultural backgrounds. Although many mental health organizations have implemented educational strategies to combat stigma, it continues to persist. To more effectively address both public and self-stigma, it is essential to incorporate social contact into educational campaigns, ensure accurate and responsible media representation of mental disorders, and expand community-based services like the clubhouse model. These community-level actions can contribute to larger systems of change that lead to better mental health outcomes and reduce stigma.

## Data Availability

The qualitative data supporting the findings of this study are not publicly available due to the sensitive nature of the information and the potential for identifying participants, even in de-identified form. Participants were assured of confidentiality and informed about the limitations of data use during the informed consent process. As such, data sharing is not possible in order to protect participant privacy and uphold ethical standards.

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
