# Peer review of "The Unspoken Struggles from Mental Health Stigma in a Rural Community: A Qualitative Exploration of Clubhouse Members’ Lived Experiences"

_ijerph, 2025, doi:10.3390/ijerph22111626_

Round 1

Reviewer 1 Report

Comments and Suggestions for Authors

Dear Authors

Considering the prevalence of mental disorders and the impact of stigma on the quality of life of mental health care users, this study is important to highlight the significant role played by Clubhouses and related organizations in curbing stigma. There are concerns that must be addressed to strengthen this manuscript.is 

Author Response

Comment 1: Very little is said about the design. The authors should provide detailed
information about the design and justification for use.

Response 1: Thank you for bringing this to our attention. We have elaborated on the study design on page 3, paragraph 1, lines 120 - 127. The new update has been highlighted in the manuscript.

“ Study Design
 A phenomenological qualitative study design was employed to explore the experiences of both the clubhouse staff and members regarding stigma and the clubhouse's role in addressing it. This design was chosen because it focuses on capturing the essence of participants’ lived experiences, allowing for an in-depth understanding of how individuals experience mental health stigma within rural communities [22]. By using phenomenology, the study aimed to develop a detailed description of staff and members’ perspectives, highlighting the clubhouse’s role in supporting individuals with mental health conditions in rural areas.”

Comment 2:  Lines 84- 87: Members were asked about their experiences, perceptions, and
behaviors within the context of their Clubhouse membership. Staff interviews followed in a similar structure but excluded questions about personal experiences with mental health and discrimination, focusing instead on the mission and support provided by the Clubhouse and how those goals are achieved. The authors explained the difference in the questions asked between staff and Clubhouse members, but this explanation comes too late. It is suggested that this explanation be included in the introduction, where the questions that the study was designed to answer are introduced. The questions asked from staff members must be clearly stipulated.

Response 2: We appreciate this suggestion. The questions for each group have been stated in the introduction section, on page 3, paragraph 8, lines 115 -117. This new statement has also been highlighted in the manuscript.

“The semi-structured interviews were tailored to each group: members' questions focused on their personal experiences with stigma, while staff focused on the Clubhouse’s mission and member support.”

Comment 3:  The interviews lasted between 15 and 75 minutes. it is common for interviews to differ in duration, but that difference is usually not significant. In this study, interviews lasting between 15 and 75 minutes raise questions. Can the authors explain the reason for this significant difference?

 Response 3: We thank the reviewer for this observation. There was one interview that lasted only 15 minutes because the member was very new to the Clubhouse, with limited experience, and was unable to speak to all the services offered. The longest interview of 75 minutes was from one of the founding members of the clubhouse so they were able to share a much richer history of the clubhouse and how it has changed over the years. We have added the following sections to the manuscript, lines 166 - 170: 

“Interview lengths ranged from 15 to 75 minutes, with variability largely reflecting participants' availability and the length of their experience with the Clubhouse environment. Newer members had limited experience and could not speak to the services offered by the Clubhouse. Long-term members spoke at length about the history of the clubhouse, the evolution of services, and changing community perspectives. ”

Additionally, another limitation has been added on lines 450 – 455.
“Limitations section: Although some interviews were shorter than anticipated, they provided focused and relevant insights. Because this was a new research initiative, some participants may have also felt hesitant to share more in-depth experiences until greater trust was established with the research team. However, the data were sufficient to achieve thematic saturation across participants, and triangulation with field notes enhanced the credibility and richness of the findings.”

Comment 4: Line 124 authors stated that participants received $100 gift for participating in this study. This is an ethical aspect that may be interpreted as deception of the participants. Were the participants informed of this gift before participating? Secondly, were they also informed that they could withdraw from participating at any time without any consequences?

Response 4: We thank the reviewer for bringing up this concern. We informed participants of the $100 gift card incentive as part of the recruitment process (via email and flyers). This amount was agreed upon with the community partner organization as an acceptable incentive for their participation. And yes, as part of informed consent, participants were informed that they could withdraw from the study at any time without consequences. Please see below for our language from the approved IRB: 

“You can choose not to take part in the research, and it will not be held against you. Choosing not to take part will involve no penalty or loss of benefit to which you are otherwise entitled. What happens if I say yes, but I change my mind later?  You can leave the research at any time, and it will not be held against you. If you stop being in the research, already collected data may not be removed from the study record.”

Here is the language we have added to the manuscript on lines 151 – 152.
“As part of the informed consent, participants were informed that they could withdraw from participating at any time without any consequences.”

Comment 5:  Line 199- technical error. A track change.

Response 5: Thank you for bringing it to our attention. It has been resolved.

Reviewer 2 Report

Comments and Suggestions for Authors

This is a straightforward study, written from a 'grassroots' perspectives. The information is general in nature, largely known and with a small sample.

It would be good to say a bit more in the introduction about the Clubhouse model and the evidence that currently exists for this model.

On page 2 line 56, the concept of 'self-stigma' seems to be dropped into the discussion of stigma without being explained properly in its own right. The authors tend to switch between stigma and self-stigma and need to tighten up the discussion of these concepts here and throughout the manuscript.

p.2 It would be useful to also explain to readers the size of Wyoming to clarify the rural issue and potential distance and isolation/social connection issues, etc. An international ready may not fully grasp the issues, and the significance of 500,000+ population may not be recognised fully without this added context regarding size of land over which that population is spread.

Please consider expanding the data analysis section of the methods section. Also, using a qualitative research reporting guide tool like COREQ would strengthen the manuscript overall. There are several criteria that are either missing or only superficially described.

p.6 CIT - explain in a [  ] bracket for readers who won't be familiar with this process of training.

p.7 line 321 'emotional regression' sounds like a very medicalised and derogatory term. I think you need to synthesis this idea further, given it seems to be drawn from one particular quote in the results section. 

The implications section is quite general and it would be good to deepen the ideas here and apply more sophistication to the rural context.

I noticed citation 34 in the reference list seems to only list author surnames (initials missing?)

Author Response

Comments 1: On page 2 line 56, the concept of 'self-stigma' seems to be dropped into the discussion of stigma without being explained properly in its own right.  The authors tend to switch between stigma and self-stigma and need to tighten up the discussion of these concepts here and throughout the manuscript. 

Response 1: Thank you for bringing this to our attention. We agree with this comment. Therefore, we have defined what self-stigma is on page 1, paragraph 2, and on lines 38 – 41. We have highlighted the changes in the manuscript as well.

“ The stigma can be either self-stigma, which involves internalized negative beliefs and blaming oneself for one's condition, or public stigma, which is a negative attitude and beliefs from individuals towards those who live with mental illness.”

Comments 2: p.2 It would be useful to also explain to readers the size of Wyoming to clarify the rural issue and potential distance and isolation/social connection issues, etc. An international ready may not fully grasp the issues, and the significance of 500,000+ population may not be recognised fully without this added context regarding size of land over which that population is spread.

Response 2: We have, accordingly, revised the paragraph to emphasize this point. 
We have included Wyoming's population density and its impact on the state's population. This update can be found on page 2, paragraph 4, and lines 75 -78. We have highlighted the changes in the manuscript.

“Wyoming, one of the most rural states in the United States, has a population density of approximately six people per square mile [17]. Residents are widely dispersed across the state, with a small population in each town, making it difficult for one to disclose or seek early mental health care [6,17]. ”

Comments 3: Please consider expanding the data analysis section of the methods section. Also, using a qualitative research reporting guide tool like COREQ would strengthen the manuscript overall. There are several criteria that are either missing or only superficially described.

Response 3: Thank you for pointing this out and recommending COREQ. We have further elaborated the data analysis and methods sections, following the guidelines of COREQ. The changes have been highlighted in the manuscript on lines 182 – 192.

“In the first phase, the two researchers familiarized themselves with the data during the transcription and cleaning process, followed by repeated readings prior to analysis. In the second phase, both researchers independently generated initial codes from the transcripts to capture meaningful patterns. Discrepancies in initial codes were resolved through team discussions led by the study supervisor, resulting in consensus and consistent interpretation among researchers. During the third and fourth phases, the researchers collaboratively reviewed and refined the codes to develop preliminary themes, ensuring that each theme accurately represented the data and aligned with the study’s objectives. In the fifth phase, the themes and subthemes were clearly defined and named to reflect their essence. Finally, in the sixth phase, the researchers synthesized all the themes to gain a clear understanding of the data and present the findings.”

Comments 4: p.6 CIT - explain in a [  ] bracket for readers who won't be familiar with this process of training. 

Response 4: Thank you for this comment. We have provided the meaning of CIT on page 7, paragraph 2, subsection 3.3.2, line 325, of the Results section. I have highlighted the changes in the manuscript.

“Crisis Intervention Team”

Comment 5: p.7 line 321 'emotional regression' sounds like a very medicalised and derogatory term. I think you need to synthesis this idea further, given it seems to be drawn from one particular quote in the results section. 

Response 5: We agree with this comment. We have explained emotional regression in simple terms on page 9, paragraph 5 of subsection 4.2 of the Discussion section, and line 390. We have highlighted the changes in the manuscript.

“ long-standing suppressed emotions”

Comment 6:  The implications section is quite general and it would be good to deepen the ideas here and apply more sophistication to the rural context.

Response 6: Thank you for your suggestion. We have elaborated on the study implications section. We have highlighted the changes in the manuscript.

Comment 7:  I noticed citation 34 in the reference list seems to only list author surnames (initials missing?)

Response 7: We appreciate pointing that out. We have updated the reference section with the correct citation, on lines 598 – 600. We have also highlighted the changes in the manuscript.

“Russell K, Rosenbaum S, Varela S, Stanton R, Barnett F. Fostering community engagement, participation and empowerment for mental health of adults living in rural communities: A systematic review [Internet]. 2023 [cited 2025 Oct 5]. Available from: https://search.informit.org/doi/epdf/10.3316/informit.993467093515364”

Reviewer 3 Report

Comments and Suggestions for Authors

This work addresses the often overlooked issue of mental health stigma within a rural American context. They have done a commendable job of grounding the study in relevant literature, clearly articulating the public health burden of mental illness, and identifying a significant gap related to the experiences of individuals in these communities. The choice to focus on the Clubhouse model as a community-based intervention is both timely and important, and the use of reflexive thematic analysis is appropriate for the research questions.

While there is much to like about this manuscript, there are several concerns that I have:

  • My biggest concern lies in a discrepancy in the study’s timeline. The authors state that participant recruitment and data collection occurred between August and September 2024. However, the Institutional Review Board approval date is listed as October 9, 2024. Conducting research with human subjects prior to receiving formal IRB approval is a serious ethical violation. This could be a simple typographical error, but as it stands, it represents a fatal flaw in the research process that must be clarified. It is impossible to endorse the publication of a study that appears to have collected data before it was ethically approved. This point requires immediate and satisfactory clarification before the manuscript can proceed.

  • The manuscript is absent of an explicit theoretical or conceptual framework. While the literature review is adequate, the study is not situated within a specific theory of stigma, recovery, or community psychology. A guiding framework would have provided a stronger conceptual foundation for the research questions, shaped the interview guide, and, most importantly, allowed for a more theoretically nuanced interpretation of the findings in the discussion. As it stands, the discussion primarily confirms existing literature rather than using the rich qualitative data to extend, challenge, or refine theoretical understandings of stigma in a rural context.

  • There are several areas where the methodological reporting could be strengthened to enhance transparency and rigor. Notably, the manuscript lacks a reflexivity statement. The authors mention engaging in reflexivity, but they do not describe their own positionality – their backgrounds, experiences, and potential biases related to mental health and rural communities – and how this may have influenced the research process from data collection to interpretation. This is a crucial component of high-quality qualitative research.

  • Additionally, the rationale for the sample size is unclear. There is no mention of whether data saturation was reached or used as a guiding principle for ending recruitment.

  • The description of the data analysis as a “template-based” approach alongside reflexive thematic analysis is ambiguous and could benefit from clarification to assure the reader of a consistent and rigorous analytic process.

  • I would also encourage the authors to reflect on the participant compensation. Providing a $100 gift card for an interview lasting between 15 and 75 minutes is quite substantial. While compensating participants for their time is essential, such a high amount for a relatively brief time commitment could introduce the risk of undue influence or coercion, particularly for individuals who may be experiencing economic hardship, a common challenge for those living with serious mental illness. A brief justification for this level of compensation or a note about it in the limitations section would strengthen the manuscript’s ethical considerations.

  • Finally, there are minor areas where the framing could be strengthened. For instance, in the introduction, the authors state that Wyoming is has a high prevalence of mental disorders and note that 31.5% of adults reported symptoms, compared to a national average of 32.3%. Citing a state prevalence that is actually lower than the national average slightly undermines the argument. The point would be more powerfully made by focusing on the state’s more alarming and distinct statistics, such as its elevated suicide rate, which is also mentioned.

While this paper presents a well-conducted qualitative study with important findings, the apparent ethical discrepancy regarding the IRB approval timeline is a serious barrier to me recommending publication at this time. Should the authors provide a satisfactory explanation or correction for this issue, I would reconsider the manuscript for publication after the authors address the aforementioned theoretical and methodological concerns.

Author Response

Comment 1:  My biggest concern lies in a discrepancy in the study’s timeline. The authors state that participant recruitment and data collection occurred between August and September 2024. However, the Institutional Review Board approval date is listed as October 9, 2024. Conducting research with human subjects prior to receiving formal IRB approval is a serious ethical violation. This could be a simple typographical error, but as it stands, it represents a fatal flaw in the research process that must be clarified. It is impossible to endorse the publication of a study that appears to have collected data before it was ethically approved. This point requires immediate and satisfactory clarification before the manuscript can proceed.

Response 1: Thank you for bringing this to our attention. There was a typo with the approval date. The correct approval date was July 30th, 2024. Interviews were collected between August and September 2024. On September 10, 2024, not October 9, the IRB approved a modification to bring on an additional graduate research assistant. We apologize for this confusion and have corrected the date within the manuscript on line 502.

Comment 2:  The manuscript is absent of an explicit theoretical or conceptual framework. While the literature review is adequate, the study is not situated within a specific theory of stigma, recovery, or community psychology. A guiding framework would have provided a stronger conceptual foundation for the research questions, shaped the interview guide, and, most importantly, allowed for a more theoretically nuanced interpretation of the findings in the discussion. As it stands, the discussion primarily confirms existing literature rather than using the rich qualitative data to extend, challenge, or refine theoretical understandings of stigma in a rural context.

Response 2: Thank you for highlighting this omission in our manuscript. We did use several conceptual frameworks, including a recovery-oriented framework and the stigma theory, as we developed and analyzed this project. We also utilized a community-engaged approach, as highlighted. We have added additional details into the introduction (lines 86 – 107), methods, and the discussion (lines 433 – 444) to further highlight these and deepen our contribution to stigma in a rural context. 

Comment 3:  There are several areas where the methodological reporting could be strengthened to enhance transparency and rigor. Notably, the manuscript lacks a reflexivity statement. The authors mention engaging in reflexivity, but they do not describe their own positionality – their backgrounds, experiences, and potential biases related to mental health and rural communities – and how this may have influenced the research process from data collection to interpretation. This is a crucial component of high-quality qualitative research.

Response 3: We appreciate bringing this to our notice. We have included the reflexivity statement in the data analysis section, and the researchers' positionality has also been disclosed on lines 196 – 207.

“Researcher Team Positionality and Reflexivity 
Our study team members’ intersecting identities and shared commitment to reducing health inequities for rural populations affected study design, interactions with study participants, and data analysis. Our perspectives include intersecting racial and ethnic identities (African American, Ghanian, and Nigerian), research experiences (career researcher and graduate students), and disciplines (sociology, public health, clinical medicine, food system and nutrition, community health, and mental and behavioral health). Both RK and GO are new to mental health research, which helped minimize potential bias during data interpretation. Throughout the analytical process, the researchers held regular meetings and engaged in reflective discussions to examine how individual research backgrounds might influence data interpretation. Additionally, we consulted with our community partner on our data analysis and interpretation. 

Comment 4:  Additionally, the rationale for the sample size is unclear. There is no mention of whether data saturation was reached or used as a guiding principle for ending recruitment.

Response 4: We agree with your comment. The rationale for the sample size is explained on page 3, paragraph 2, in the study settings and participants section, lines 136–140. The changes have been highlighted in the manuscript as well.

“Recruitment concluded after 16 participants, as no additional volunteers were found. Additionally, subsequent interviews after our 10th interview revealed similar findings, indicating that data saturation had been reached. Based on this, the research team determined that sufficient data had been collected to explore the study objectives.”

Comment 5:  The description of the data analysis as a “template-based” approach alongside reflexive thematic analysis is ambiguous and could benefit from clarification to assure the reader of a consistent and rigorous analytic process.

Response 5: Thank you for bringing this to our notice. We have elaborated on the reflexive thematic analysis and removed the ‘template-based’ approach to clarify the analytic process, on pages 4, lines 182-192. The revision has also been highlighted in the manuscript.

“ In the first phase, the two researchers familiarized themselves with the data during the transcription and cleaning process, followed by repeated readings prior to analysis. In the second phase, both researchers independently generated initial codes from the transcripts to capture meaningful patterns. Discrepancies in initial codes were resolved through team discussions led by the study supervisor, resulting in consensus and consistent interpretation among researchers. During the third and fourth phases, the researchers collaboratively reviewed and refined the codes to develop preliminary themes, ensuring that each theme accurately represented the data and aligned with the study’s objectives. In the fifth phase, the themes and subthemes were clearly defined and named to reflect their essence. Finally, in the sixth phase, the researchers synthesized all the themes to gain a clear understanding of the data and present the findings.”

Comment 6:  I would also encourage the authors to reflect on the participant compensation. Providing a $100 gift card for an interview lasting between 15 and 75 minutes is quite substantial. While compensating participants for their time is essential, such a high amount for a relatively brief time commitment could introduce the risk of undue influence or coercion, particularly for individuals who may be experiencing economic hardship, a common challenge for those living with serious mental illness. A brief justification for this level of compensation or a note about it in the limitations section would strengthen the manuscript’s ethical considerations.

Response 6: We thank the reviewer for sharing this concern. The incentive amount was agreed upon with the community partner organization as an acceptable incentive for their participation. We used a community-engaged research approach for this project. Our community partner wanted the voices of their clubhouse members heard in the larger discussions about mental health in their community. We worked with the community partner to develop the project, including the question guides, the incentive payment amounts, and the recruitment process. We continue to engage them in the data analysis and interpretation phase, as well as in disseminating the findings and outlining the organization's next steps. 

Comment 7:  Finally, there are minor areas where the framing could be strengthened. For instance, in the introduction, the authors state that Wyoming is has a high prevalence of mental disorders and note that 31.5% of adults reported symptoms, compared to a national average of 32.3%. Citing a state prevalence that is actually lower than the national average slightly undermines the argument. The point would be more powerfully made by focusing on the state’s more alarming and distinct statistics, such as its elevated suicide rate, which is also mentioned.

Response 7: Thank you for this suggestion. We have highlighted on Wyoming suicide rate statistics on page 2, paragraph 4, and lines 82 and 83. The changes have been highlighted in the manuscript as well

“increased slightly to 157 in 2023, corresponding to an age-adjusted rate of 26.3 per 100,000 people, ranking Wyoming as the state with the third-highest suicide rate ”

While this paper presents a well-conducted qualitative study with important findings, the apparent ethical discrepancy regarding the IRB approval timeline is a serious barrier to me recommending publication at this time. Should the authors provide a satisfactory explanation or correction for this issue, I would reconsider the manuscript for publication after the authors address the aforementioned theoretical and methodological concerns.

Round 2

Reviewer 2 Report

Comments and Suggestions for Authors

The revisions are excellent. The authors have done well to add further COREQ-informed information to the methods section, and other revisions are good too.

Reviewer 3 Report

Comments and Suggestions for Authors

Overall, the authors have done an excellent job of addressing all concerns I have raised in the initial review. The revisions are thorough, thoughtful, and have substantially strengthened the manuscript.